# Automobilities after SARS-CoV-2: A Socio-Technical Perspective

**Liqiao Wang \* and Peter Wells \***

Business School, Cardiff University, Cardiff CF10 3EU, UK
* Correspondence: WangL45@cardiff.ac.uk (L.W.); WellsPE@cardiff.ac.uk (P.W.);
  Tel.: +44-(0)7-9220-6718-3 (L.W.); +44-(0)7-8096-9606-2 (P.W.)

**Abstract:** This paper presents an analysis, informed by socio-technical transitions theory and the socially derived concept of automobility, of the impact of the SARS-CoV-2 virus and resulting COVID-19 pandemic on automobility in Europe. The paper argues that the concept of a pervasive, sudden, and powerful crisis has not previously be explored in the socio-technical transitions literature. The strong behavioural changes in physical and virtual mobility associated with the pandemic are argued to be particularly significant, representing a 'living lab' in which to explore the possibilities for disintegrating the boundaries of the automobility system, thereby breaking the enduring structures and practices that have enabled automobility to remain largely unchallenged in the policy arena. Change processes previously underway in the automotive industry and in automobility are not impacted equally by the pandemic. We present initial evidence that mobility sharing will reduce, while the acceptance of electric cars will increase. However, it is also concluded that the hegemony of private automobility is not in itself threatened by pandemic outcomes.

**Keywords:** coronavirus; pandemic; automobility; Europe; crisis events; futures

---

## 1. Introduction

It is now widely accepted that the emergence of the SARS-CoV-2 virus has triggered an unprecedented global crisis. The severity, geographic scope, and pace of emergence of this health crisis has in turn instigated a series of repercussions that will probably take years to become fully apparent. Thus, the health crisis has in turn triggered economic crises, and then emergent socio-political crises [1–4], with widespread public debate over how the world will be changed as a result [5,6].

It is further apparent that the nature and severity of the impact of the pandemic has varied immensely. Distinct socio-economic classes, and specific places, have suffered much more than others in ways that are not yet fully understood. Unsurprisingly, major international transport hubs such as Paris, London, and New York have featured as early 'hot spots' for the spread of the virus.

Mobility has been at the heart of the pandemic crisis, having a key role in the transmission of the virus from the outset via international air travel, and then via mass public transit systems. Moreover, the cessation of mobility has been at the centre of the policy response, generically known as 'lockdown'; and ultimately, mobility is being positioned as a central feature of many policy initiatives to restart national economies and as an indicator of some sort of return to a 'new normal'.

Physical mobility is therefore woven tightly into the pandemic narrative, and within that, the interplay of different transport modes is once again seen as crucial both to controlling the spread of the pandemic and to socio-economic recovery thereafter. Virtual mobility has come to unprecedented primacy in these circumstances [7] although it is also apparent that much of traditional physical mobility is a vital component of contemporary economies. Moreover, the pandemic has been identified

as providing an opportunity to challenge the primacy of business, economics, and market forces, and rather emphasise policies that nurture sustainability in a broad sense [8].

In this paper, we explore the potential futures of automobility using the lens of socio-technical transitions theory. We focus on the situation in Europe (taken to mean the European Union, the European Free Trade Area, and the United Kingdom). While empirical evidence on the impact of the pandemic on automobility is limited to date, we outline some key areas of contestation over future automobilities, and some emergent outcomes. We argue that the pandemic has the potential to influence regime transition pathways, in part at least through the re-orientation of diverse 'frames' through which interest groups define automobility [9]. That is, transition pathways may be slowed down, accelerated, stalled, or given a different trajectory as the pandemic and its attendant impacts reverberate through societies. In this, we focus on the character of system boundaries under different transition conditions. We also propose a hierarchy of impacts in which automobility is a subset of all mobilities, and that automobility in turn can take a variety of forms.

Section 2 provides an account of how a crisis event might be integrated into socio-technical transitions theory, and its application to automobility. We argue that the pandemic has created real-time events and data on behavioural change and policy interventions that would not have been possible to imagine previously. In Section 3, we provide an account of automobility before and after (or at least in the immediate aftermath of) the crisis to argue that there are countervailing or contradictory pressures between the 'sustainable mobility' and 'economic recovery' positions, but the crisis nonetheless represents a crucial moment in the disintegration of the integrity of the automobility socio-technical system. Section 4 then summarises the implications for future research in transport, mobility, and socio-technical systems.

## 2. Socio-Technical Transitions and the Significance of Crisis

Socio-technical systems are comprised of coherent networks of production and consumption in which the key constituent elements are largely self-reinforcing. Systems in this sense are distinct from 'hard' or 'closed' systems as understood in the physical sciences. Rather, they are user-defined, albeit grounded in some concrete reality. At the core of socio-technical systems is a 'regime' of dominant technologies and associated social or economic structures. These structures include markets, companies, labour organisations, prevailing social norms and behaviours, legal frameworks, and government, which act collectively to provide coherence to the system [10,11]. Socio-economic institutional entities are in this conceptualization also system components that mediate the agency of actors, and as such are intimately involved in the definition of system structure and change [11]. The domain of socio-technical transitions is then the focus of research and policy into enabling the transition of an existing socio-technical system into one that is more sustainable along one or more dimensions. In a sense, the concept of systems used in this domain of research is an abstraction and simplification, or a metaphor as a means of assisting in the visualization of important but complex processes of change in society. Metaphors drawn from the physical sciences are often used in this manner, for example in the fields of evolutionary economics or of industrial ecology [12,13].

That is, theoretically, a socio-technical system has two important properties: It is reproduced by the internal relationships of the system components, and it has a defined conceptual boundary that can be used to confine the unit of analysis. The boundaries within and between systems are not entirely fixed but have become of research interest as previously distinct socio-technical systems have come to coalesce in multi-system interactions [14,15]. These two properties convey a dynamic stability to the socio-technical system. It is therefore neither static nor entirely fluid. The socio-technical system is generally held to be in a state of dynamic stability rather than a more neo-classical state of equilibrium. In this state of dynamic stability, the pace and direction of system change are mediated by system incumbents and the processes of institutionalisation via for example cultural and behavioural norms, regulatory constraints, and a broadly shared vision of what is, and is not, possible.

Generally, in socio-technical system research the focus is on sustainable transitions [16]. The primary questions are why does change happen, and how does change happen? In seeking to understand and find answers to these questions, research has been centred on the historical analysis of changes to socio-technical systems, in terms of 'bottom up' emergence of niches premised on technological innovations and novel behaviours [17,18], or on 'top down' pressures (sometimes termed the 'landscape level') from events that are external to the system but have a bearing upon it. Inevitably, the major systems of production and consumption have been the locus for the most research, including transportation, energy, agriculture, water, and urbanism [16,19]. A growing area of interest has been on how existing regime structures may become destabilised, and thereby create the 'space' for more sustainable practices to emerge [20–22].

Previous research has also identified the importance of dominant incumbents in maintaining regime structures [23,24]. New entrants, often identified as small and entrepreneurial companies, have long been recognised as potential regime disrupting entities [25,26]. More recently, attention has turned to the ways in which both these assumptions may not always hold. That is, incumbents can themselves be an important source of regime change because they have the scale, assets, and competencies to make a difference, and new entrants can be large, mature companies 'migrating' from their traditional realm of operations. Moreover, transition pathways [27] may be strongly influenced by policy or regulatory changes that frame market conditions [28,29].

The role of a landscape 'crisis' event as a decisive historical moment that transforms a socio-technical system is not well developed in the socio-technical transitions network (STRN) literature, where the focus has been on the gradual unfolding of transition processes [30,31]. Again, some research has sought to understand how transitions might be accelerated [32], in part because of the perception that sustainability problems were still growing, and planetary boundaries were in danger of being breached despite decades of policy intervention [33]. In essence, economic growth and population expansion, fuelling consumption growth, have continued to overwhelm eco-efficiency improvements.

However, the pandemic event is clearly different from the 'climate crisis' around which the scientists of the Intergovernmental Panel on Climate Change (IPCC) have been seeking to orchestrate planned reductions in carbon emissions. By virtue of the characteristics noted above, the pandemic stimulated rapid and profound policy measures with a remarkable degree of conformity around the world—notwithstanding some outliers such as Brazil and the United States. In parallel, citizens collaborated with the restrictions despite the personal emotional and financial hardships that ensued. That is, the pandemic demonstrated that large-scale behavioural change was possible, albeit for a period of weeks only. Again, the focus of transitions research has mostly been on technological innovation as the primary trigger to initiate system change [34], around which behaviours and attitudes, infrastructures, and institutions, might change with time. However, it has been recognised that the 'lens' of socio-technical transitions can contribute important insights into understanding the impact of the pandemic crisis event [30,31]. In part, the contribution of socio-technical transitions theory draws on the insights from research into the history of previous transitions in which a constellation of agents has acted as participants in system change processes wherein technological diffusion arises out of the co-construction of a technology in its contextual setting [35].

To simplify this discussion, Table 1 summarises three main conditional states for a socio-technical system and does so for three hierarchal levels. The three main conditional states are dynamic stability, normal transition, and crisis. Dynamic stability exists where change processes are constrained within the socio-technical system, as could be argued to be the case for the automotive industry up to the 2020s. For companies in the regime, this means the normal processes of competition apply. Companies prosper based on either cost leadership or product performance differentiation, and typically over time there is a consolidation at the industry level tending towards oligarchy. The stable reproduction of the regime is enhanced by continued revenue growth within existing markets and by the geographic expansion of markets. Conceptually, this means that the socio-technical system can be considered largely self-contained, and that the boundaries are impermeable. This means that there are limited

interactions with other (adjacent) socio-technical systems, and new entrants are unable to penetrate the regime. A degree of technological and socio-economic convergence, particularly around digitization and electrification, has resulted an erosion in this boundary condition for systems that used to be considered separate. In multiple ways, there are thus varying degrees of 'overlap' between say automobility, housing, electricity, food, telecommunications, and so forth.

**Table 1.** Overview of socio-technical system change at different levels.

| System Level | | Patterns of Change | | |
|---|---|---|---|---|
| | | Dynamic Stability | Transition | Crisis |
| Corporate Innovation Strategy | Micro | Business as usual/normal competition | Competing differently/competing better | Beyond competition |
| Regime | Meso | Regime reproduction | Regime reorganisation/Regime amidst diversification | Magnified transition pathways |
| System Boundary Conditions | Macro | Impermeable | Porous/Fractured | Disintegrating |

(Source: Authors).

Figure 1 illustrates schematically the impact of a crisis event on socio-technical transitions. Reading from left to right, the socio-technical system is assumed to be in a state of dynamic stability ahead of the crisis event. There is, however, also an assumption that alternative technologies, behaviours, and practices are potentially available, while adjacent socio-technical systems are potentially developing towards the focal system. Hence, incipient transition pathways may be present. The crisis event then has the potential to accelerate or magnify change processes. First, it weakens the existing socio-technical regime. Markets may collapse, dominant incumbents are no longer profitable, and behaviours and practices can change rapidly. In parallel, the crisis may result in a failure of legitimacy for the dominant socio-technical system, precipitating a shift in government policies and a new openness to alternatives. Three broad outcomes are portrayed. The socio-technical system may become destabilised and collapse, it may recover and resume on a pathway of dynamic stability, or it may undergo an accelerated transition to a new socio-technical system.

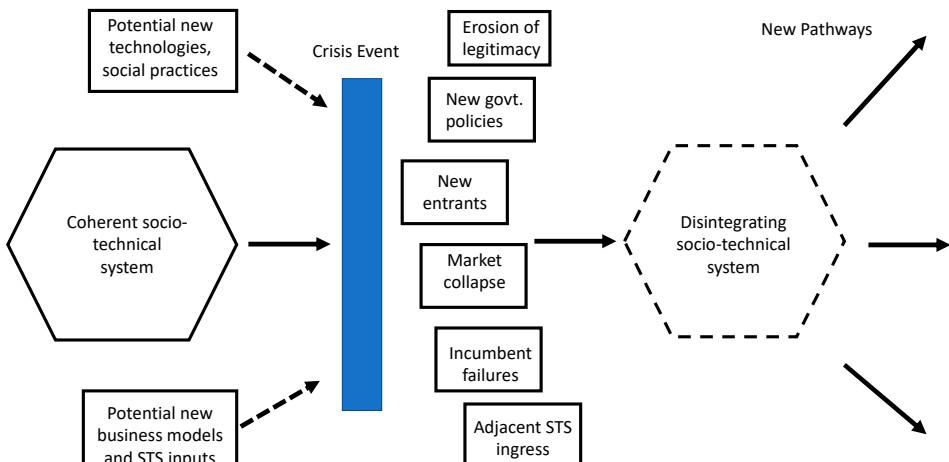

**Figure 1.** Crisis events and socio-technical transitions.

Under conditions of transition, there are several pathways possible [27,36] under which the regime might be reconstituted. Transition processes might be instigated by companies acting at the micro level, via new technology introductions (competing better) or via business model innovation (competing differently) to change the terms of competition [37]. Equally, the ability of the system to contain landscape level pressures might be exceeded, resulting in fractures in the regime (defined as distinct break points in the system boundary) or a more generalised porous system boundary wherein

boundary edges become less distinct over time. Again, the boundary conditions defined in this paper can be considered as metaphors for complex societal interactions and processes that are both contested and uncertain in outcome.

The conditions of meta-system crisis have not been subject to research thus far. However, as the circumstances of crisis are acute, widespread, and almost instantaneous it can be expected that there will be the potential for profound system impacts. System integrity under these circumstances may be derived from one of two sources. Either the system is of sufficient flexibility to 'bend' under the pressure of events and then return to stability; or the system is effectively insulated from external events, even one of the magnitude of the coronavirus pandemic, and hence endures in the pre-existing condition. However, it is to be expected that most socio-technical systems will be affected by the crisis event. At the corporate level, the rapidly changed conditions might enable radically different forms of competition that combine innovative technologies and new business models premised on greatly enhanced sustainability. At the regime level, we hypothesise that a crisis event will magnify and accelerate pre-existing regime trends. In some cases, this can lead to the system boundaries disintegrating as the socio-technical system collapses, being simply untenable in the post-crisis era.

Such impacts may or may not be mitigated by regime stabilisation measures enacted by governments, or other institutional arrangements. There may well be a crisis of legitimacy for the 'old' way of doing things, including for the previous accepted political structures that may be seen to have 'failed' in the crisis event. There will also be a renewed and more urgent questioning of the mundane reality of life given that the pandemic has shown how alternative arrangements are indeed viable. It can be hypothesised, then, that the immediate aftermath of the crisis event will be a renewed level of contestation over the future in a broad sense. In terms of transport and automobility, there is great uncertainty over the future of business air travel, regular commuting by car or by public transport, working from home, Internet shopping and home delivery, holidays, and travel to participate in major public events such as football matches or music concerts.

Prior to the emergence of the pandemic, research had already identified that reaching stated climate change goals via emissions reductions was unlikely to be achieved via technological innovation alone, or even by choice of transport mode. Equally, lifestyle change alone, without eco-efficiency measures and optimisation of mode choice, is unlikely to be sufficient [38,39]. Researchers had therefore been arguing for a greater focus on addressing the needs, preferences, experiences, and identities of users. The pandemic thereby provides a 'living lab social experiment' on the ability to change the apparently stable and immutable practices of mobilities, which policy makers have largely regarded as sacrosanct [40]. People have been compelled to change mobility behaviours in ways that were unimaginable to transport policy makers in the past. While in the past there have been ethical concerns raised over social science experiments, the SARS-CoV-2 virus has presented researchers with a wealth of real-time events and data, in which to explore many aspects of human behaviour [41].

In terms of Table 1, it can be postulated that the pandemic may indeed precipitate the disintegration of the socio-technical system boundaries, which will be key to deconstructing the constituent features of the 'car dependent transportation system' identified by Matiolli et al. [42].

## 3. The Automobility Socio-Technical Regime: Before and After

As noted above, we can expect socio-technical system change arising from the pandemic, but equally there will be an element of magnification and acceleration of pre-existing trends. Automobility in Europe was subject to these pre-existing trends. This section reviews in Section 3.1 the character of automobility up to the end of 2019, based upon literature searches and the reflexive expertise of the authors. The use of secondary data was mandated by the situation, in which we sought to understand the impact of COVID 19 as events were unfolding and being reported. In addition, the scope of the discussion is very broad. We take in multiple aspects of ongoing change in automobility using a range of sources including the mainstream academic literature, and reports or other data from governments, agencies, companies, consultants, and others. It is not possible to cover this broad range though

primary research. Rather, we adopt the methodological approach of longitudinal immersion [43]. Section 3.2 then considers the period since the emergence of the pandemic to the present (mid-2020), and inevitably largely draws on contemporary news accounts to capture developments in policy, industry strategy, and related matters.

### 3.1. Automobility up to the End of 2019

At the highest level, the institutions of globalisation and political-economic integration had been under pressure for some time, by those proposing a return to more nationalist orientations. Before the pandemic emerged, there was already a strand of discussion that was arguing that globalisation was failing to resolve social, environmental, and economic dysfunctionality [44,45] and resulted in flawed corporate strategy [46]. Manifestations of these trends include the decision to leave the EU by the UK, the attacks by US President Trump on the IPCC, World Trade Organisation, North America Free Trade Area, World Health Organisation, and North Atlantic Treaty Organisation, and the growing popularity of nationalist/separatist political parties in Europe. When the pandemic emerged, those nationalist preoccupations became more evident with the closing of national boundaries (e.g., in the Schengen Zone), and in the difficulties of orchestrating an agreed pan-European Union response. In turn, countries in Europe have enacted widely different policy responses with respect to the automotive industry and to automobility in general.

Similarly, there has been a long-running tension between the desire to retain a sizeable automotive industry, and achieving national and urban targets on carbon emissions reductions, improved air quality, and other environmental concerns [47,48]. There were considerable expectations regarding the performance of new technologies and practices for automobility, along with some concerns, ahead of the pandemic, as is summarised in Table 2. Four aspects of innovation are identified, of which three are grounded in technological innovation and one in social or organisational innovation. Connectivity refers to the processes of digitisation and communication that are ongoing across many social and economic realms, including automobility. Autonomous driving is more specific to the realm of transport and, especially, automobility. Autonomous driving has been widely applied in more constrained transport systems (e.g., aircraft, rail) where there are predictable and stable parameters, but is more an emergent area of application in cars. Similarly, electrification of various transport modes has been ongoing for some time, but to date, remains a niche application in cars. Finally, sharing is a social and organisational innovation (albeit one that has been significantly enabled by smartphone connectivity) that is promoted to reduce the environmental burdens of cars by behavioural change rather than technology innovation.

Of the four main technologies and practices identified in Table 2, the pandemic has been of greater significance to shared vehicles and to fleet electrification, as is discussed more fully below. The pace of deployment of connectivity technologies and those to enable autonomous vehicles may be slowed by the wider economic impact of the pandemic in undermining the financial health of the industry.

Shared vehicle fleets were becoming of more significance prior to emergence of the pandemic, in the form of ride hailing (e.g., UBER), ride sharing (e.g., BlaBlaCar), and many forms of car sharing (e.g., Greenwheels). However, shared vehicles comprised a residual and very minor share of the new car market, even in Europe where such schemes were best established. Moreover, there had been some high-profile failures (e.g., Autolib in Paris), while the private sector initiatives of BMW and Mercedes were merged in an attempt to gain scale. In short, vehicle sharing was a recognised niche and a potential component in future mobility but was weakly established and suffered notable setbacks. In contrast, shared mobility in the form of scooters and bicycles (especially electric versions) was undergoing rapid (and sometimes chaotic) expansion in many cities in Europe.

**Table 2.** Anticipated issues and benefits from connected, autonomous, shared, and electric vehicles.

| Item | Connected | Autonomous | Shared | Electric |
|---|---|---|---|---|
| Carbon emissions | Cars connected to infrastructure management should reduce total emissions. | More efficient driving should reduce emissions per car BUT growth in demand could increase total overall | Carbon emissions should be less if shared vehicles result in less traffic. | Zero emissions from the vehicle, otherwise depends upon electricity supply. |
| Road deaths and injuries | Should be reduced if connected to infrastructure management. | Should be reduced significantly BUT some concerns over non-car road users. | Some potential reduction if there is less traffic overall. | Greater mass could increase severity of collisions. |
| Air quality | Could be reduced e.g., by geo-fencing urban areas. | Could be reduced by more efficient vehicle use | Some potential improvement if there is less traffic overall | Significant improvement as zero emissions at point of use. |
| Noise | No significant change. | Per vehicle there is no improvement. | Some potential improvement if there is less traffic overall | Significant improvement, especially at low speeds. |
| Congestion | Significant reductions in many sources of congestion expected. | Should improve capacity utilisation and reduce congestion due to crashes. | Some potential improvement if there is less traffic overall | No significant impact |
| Environmental burdens of infrastructure | Substantial investments in hardware needed. | Some further investment in infrastructure likely | No additional environmental burdens | Charging station networks require investment. |
| Life Cycle Analysis | Rare metals inside sensors, etc. Also, copper for telecommunications, energy for server farms. | Metals for multiple electronic components | No additional environmental burdens | Large amount rare earths, cobalt, copper, and lithium required |

(Source: Authors).

Electrification of the fleet was becoming more significant up to the end of 2019 in many markets. Following the pioneering efforts of companies like Nissan, Renault, BMW, and Tesla, from around 2012 onward there was a growing support from governments in fiscal and other incentives, and in infrastructure investments. Following the diesel emissions scandal in 2015, VW Group underwent a strategic reorientation towards electric vehicles, centred initially on the MEB matrix vehicle concept. With declining battery costs, improved range, and a broader choice of brands and models, the market for battery electric vehicles was expanding rapidly as the industry entered 2020, with much of the discussion centred on shortage of supply (of batteries) restraining sales.

*3.2. Automobility in 2020*

New car sales have not returned rapidly to pre-pandemic levels. In many countries in Europe, sales of new and used cars fell to almost zero for a period of 6–8 weeks as lockdowns and workplace closures (e.g., including franchised car dealerships) took effect [49]. Annualised impacts are expected to be severe in most markets and for most manufacturers. There is a widespread expectation that many countries also face a period of austerity, with higher levels of unemployment and reduced incomes. Consumers are facing greater uncertainty than has been previously experienced and will probably be reluctant to accept credit offerings to finance vehicle purchases—a mechanism that has been key to sustaining demand over recent years.

While automobility was therefore initially a 'victim' of the effects of the pandemic when citizens were largely confined to their homes [50], the future prospects for car-based mobility have arguably been significantly enhanced by a landscape event that might act to stabilise the regime. Alternatively, many citizens experienced what many scientists have started to document: That the cessation of automobility brought visible evidence of air quality improvements. Along with the fall in carbon emissions that has also been noted, the pandemic strengthened the case that the prevailing form of automobility was a key component of global environmental problems. Many of these issues have been recorded by the mainstream media [51], but also scientific sources have started to contribute insights [52].

There are two elements to this stabilisation. First, notwithstanding potential declines in overall mobility and travel arising from changed economic circumstances and travel behaviours, the pandemic has exposed mass public transit as particularly problematic. A UK survey report on 22nd April by

Auto Trader claimed that 48% of existing public transport users would be less likely to do so once the pandemic had passed, and 68% of young people (16–24 years old) thought so [53]. In addition, over half (56%) of individuals that had a driving licence but did not have a car thought that COVID 19 would now make them consider owning a car [53]. Second, the automotive industry is still seen as one of the key pillars of many national economies, and so multiple support measures have been established to stimulate the market for new cars and to provide financial liquidity to the manufacturers.

However, the character of the stabilisation offered represents the somewhat conflicting aims of economic renewal against more sustainable mobility. At one extreme is the 'pure' economic recovery position, in which it is argued that now is not the time to pursue unprofitable 'green' technologies. Rather, the market should be bolstered to give vehicle manufacturers the revenues they need to survive this challenging period, and to reduce the cost of lockdown measures [54]. In this view, it is necessary to save the industry first and foremost, because without viable companies there will be no 'green' automobility in the future. At the other extreme is the 'green mobility' position in which it is argued that now is the time to establish active travel as the default mode, and that resources spent to allow vehicle manufacturers to keep on producing petrol and diesel cars is counterproductive. In this view, it is preferable to invest in changing infrastructures to allow active travel and to 'future-proof' public transit, while also focusing on measures to reduce the overall demand for mobility.

The first position was expressed succinctly by Eric-Mark Huitema, Director General of ACEA (the industry representative body in Europe):

> "Firstly, to take concrete measures to avoid irreversible and fundamental damage to the sector with a permanent loss of jobs, capacity, innovation and research capability. Secondly, Europe should prepare to stimulate the recovery of our sector, which will be a key contributor to the accelerated recovery of the European economy at large" [55].

The second position is advocated by groups such as the UK entity SUSTRANS. The organisation has been promoting the retention of this active travel beyond the duration of the pandemic, while the London Cycling Campaign has advocated a series of steps from short-term response to long-term strategy (see https://lcc.org.uk/articles/cycling-and-the-covid-19-crisis). Cities such as Oakland (US), Bogota, Berlin, and Vancouver have allocated lanes on multi-lane roads for pedestrian and cycle use [56]. Often, these measures are 'temporary', but they may become permanent.

While the EU has eventually orchestrated a pan-European economic stimulus package, and while individual countries have announced related macro-economic policies national, measures for the automotive industry have been quite varied. There has been discussion of a pan-European package for the automotive industry, but no actual policy announced. Table 3 provides an overview of policies on battery electric vehicles (BEVs) announced by July 2020 for selected countries in Europe.

Table 3 gives an indication of the diversity of responses to the pandemic in terms of electric car support and related measures on active travel. At one extreme, the UK has to date rejected calls for industry subsidy, bail out, or scrappage incentive. A part of the rationale is that the UK imports over 90% of cars sold, and so incentives would only assist non-UK manufacturing. In addition, Jaguar Land Rover was deemed insufficiently solvent to qualify for a government grant, though the company has raised £1 billion in financing from Chinese sources. At the other extreme, France has combined straightforward bailout support for the major manufacturers with a series of innovative policies for retrofitting internal combustion engines with batteries, and to support used BEV car purchases. Renault has promised to undertake a drastic restructuring scheme involving plant closures and redundancies, with the inevitable worker protests. Germany has resisted pressure from the industry to give more generic support such as scrappage incentives for new petrol cars but has rather concentrated any stimulus efforts on the battery electric and fuel cell vehicles. The Netherlands, with only a modest vehicle manufacturing sector to be concerned with and a well-developed active travel culture, has a focus on ensuring social equity in the support of BEV purchases that are not fundamentally seen as a means of stimulating the economy.

**Table 3.** The main policy initiatives of leading battery electric vehicle (BEV) markets in Europe along key dimensions.

| Item | UK | Germany | France | Netherlands | EU |
|---|---|---|---|---|---|
| Purchase support | Yes, pre-existing grants for BEV purchase | Yes, increased after pandemic | Yes, increased after pandemic | Yes | Possible, under discussion |
| Constraints on support e.g., price cap; income cap | Yes | Yes | Yes, price and income capped | Yes | Not known |
| Used car support | No | No | Yes | Yes | No |
| Charger support | Yes, pre-existing grants for installation | Not known | Yes | Not known | |
| Public charger support | Yes, limited pre-existing support | Yes | Yes, expanded support | Yes | Yes, substantial programme planned |
| Scrappage incentive | No | No | Yes, new scheme | No | No |
| Retrofit support | No | No | Yes, new scheme | No | No |
| Fiscal support e.g., VAT changes | No | Yes | No | Yes, for leased cars | Possible |
| R&D support | Yes, pre-existing grants | Yes | Yes, expanded after pandemic | No | Yes, pre-existing grants plus new measures |
| Support for micro-mobility | Yes, some scooter schemes supported | No | No | No | Not known |
| Active travel support | Yes, grants to convert roads to cycle use | Yes, at local level | Yes, at local level | No | Not known. |
| Bail out support | No except furlough scheme | No | Yes, for Renault and PSA | No | No |

(Source: [57–65]).

Sales of new cars by the automotive industry in Europe virtually collapsed as the pandemic took hold over March, April, and May. Markets such as Italy, Spain, and the UK saw sales in April 97% less than the equivalent month in 2019. April was the first full month that lockdown was in place over most European countries. Overall, the first quarter of 2020 resulted in sales of new cars across Europe falling by around 26% to just over three million units. Electric vehicle sales of all types, including plug-in hybrids, actually gained ground: In the first quarter of 2020, the market share was 7.5%, compared with just over 3% for the equivalent period in 2019. BEV sales (plus fuel cell vehicles) were 130,297 (up 58.2%), giving a 4.27% market share [66].

The impact of the pandemic on shared automobility is less well documented. A priori the expectation would be that such services would also be regarded as suspect by users, because of potential contamination from other users, and of course a risk to drivers. Additionally, with the lockdown measures in place in key markets, the demand for car sharing services will have fallen significantly, and with no certainty as to how quickly that demand will return. UBER announced in May that demand for rides had fallen by 80%. In the US, it is notable that Lyft signed an agreement with Amazon to undertake parcel delivery. Lyft had suffered a fall in demand for passenger rides, and therefore had drivers unable to earn income. Amazon had a surge in demand from home shopping and had a shortage of delivery capacity. It is also notable that GM shut down its Maven subsidiary, which in any case had been struggling ahead of the pandemic.

UBER has sought to capitalise on its UBER Eats business in some markets, delivering meals from restaurants to home-confined consumers. However, in May 2020 UBER announced plans to make 3700 staff redundant amid widespread restructuring that included selling off the Jump scooter ride-sharing business. UBER also announced operational changes such as limiting rides in UBER X vehicles to three people rather than four, with no passengers in the front, and with all vehicle occupants wearing masks [67]. It remains to be seen whether UBER and other ride hailing, ride sharing, and car sharing operations can survive the crisis period, but they do not feature in any rescue or recovery policies to date.

## 4. Discussion

This brief analysis has shown that socio-technical systems theory is a useful way of framing complex change processes that are more than just economic or technological in character. Behind the different outcomes shown in Table 3 lie multiple differences between the countries concerned that are expressed as (geographic) variations of the automobility socio-technical system. Given their divergent starting points in terms of industrial base, mobility infrastructure, cultures of mobility, and economic condition, the analysis shows that the pandemic is likely to exacerbate such divergence further still.

In each country, constituencies of interest have entered into the discourse with varying degrees of success. In Germany, for example, the automotive industry represented by the leading vehicle manufacturers and by the lobby group (the VDA) failed to convince the government that a broader bailout was warranted. In contrast, multiple 'green' political groupings sought to use the moment to propel the automotive industry into the electric mobility phase.

What is less clear from the analysis is the extent to which physical mobility per se will be permanently reduced, or how far active travel and micro-mobility will emerge as a substitute for both automobility and public transport. Mobility has been crucial to social integration, allowing interlocking spheres of social activity over unprecedented spatial scales [68]. Initially at least, it would be reasonable to conclude that automobility per se, maybe in electric form, has been reinforced by the pandemic given the problems faced by public transport and by the relatively fixed nature of the spatial structures of mobility around which people have built their lives.

The isolationism of the lockdown period is likely to induce multiple psychological and practical problems but reinventing the spatial structure of social lives will equally likely to be a lengthy process. The pandemic has therefore highlighted that motility is structured by place, and by the inequalities in

society [69]—and it is likely that social acquiescence to such inequalities will be less forthcoming than before the pandemic.

Moreover, transitions processes require time for their full impact to become apparent. In the early historical cases such as the transition from sail to steam, this time period could be measured in decades. In contrast, the application of socio-technical systems theory here is concerned with the very immediate impacts where only initial 'directions of travel' can be identified. Rather than looking back at the past, this paper is more about contemplating the future. There is considerable scope for further research using socio-technical transitions as the template to interpret ongoing events, even when they are of the magnitude of the coronavirus pandemic.

**Author Contributions:** Conceptualization, L.W. and P.W.; methodology, L.W. and P.W.; writing—original draft preparation, P.W.; writing—review and editing, P.W. and L.W. All authors have read and agreed to the published version of the manuscript.

**Funding:** This research has no external funding.

**Conflicts of Interest:** The authors declare no conflict of interest.

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
