# Peer review of "Automobilities after SARS-CoV-2: A Socio-Technical Perspective"

_sustainability, doi:10.3390/su12155978_

Round 1

Reviewer 1 Report

The paper is interesting and I enjoyed reading it. The discussion relating to possible future developments in automotivity is topical and worthwhile. The study uses secondary data to good effect, although they have not really explained why this was the best choice of approach. The material used in the analysis is clearly set out and displayed.

Having said this, there are a number of areas on which I have concerns. Coming from a Systems background, I found the authors' use of Systems terms confusing. When siting work in a Systems paradigm, it is important to explain which systemic perspective is taken. The authors appear to have a hard, closed systems approach similar to that taken by, for instance, classical economics. However, even within such a perspective, it is necessary to recognize that boundaries are not given, but set by the person to whom the analysis is interesting. Boundaries do not 'disintegrate' as if they were a part of the physical world. They may become inappropriate and are redrawn. Similarly, the concept of a socio-technical system does not have one, universally-agreed definition for all purposes. The concept is used across many fields, e.g. business, geography, information systems, engineering and is defined by researchers in many different ways. The authors have given a definition but without setting the context within which they believe it to be valid, nor do they provide any references to confirm its acceptance and usefulness Lines 67-76). Furthermore, there is little material in this discourse that relates to systemic analysis or STS. The authors discuss transitions, mentioning particular systems at times. Perhaps systemic analysis is not necessary to their purpose. If that is the case, what is the relevant framework and literature?

There are some sweeping statements in the Introduction that require evidence to support them. For example, on Line 77 "Generally, in STS research, the focus is on transition". Is this so? What is the field within which this is statement claims validity? Where are the references to support it? How are the authors defining 'transition'? Such references as there are in this section relate to sustainability, but not specifically to STS.

"The role of a landscape ‘crisis’ event ... is not well developed in the literature". The literature of which field? Sustainability? Socio-technical Systems? What about the literature of Risk and Crisis Management? "more recent research has sought to understand ... " More recent than what? There is only one reference in this section and it relates to Energy research.

I would like to see the authors reconsider what their objectives are in writing this paper and what is the real frame of reference for the research. A clearer statement of perspectives, including which bodies of theory are relevant, would be useful to the reader. The authors need to ensure that they have provided a complete set of references. 

While the paper is generally well-written, it would benefit from another proof-reading. In particular, in Line 138, the word "on" should probably read "one"; on Line 159, the word "address" should probably be "addressing"; and at Line 163, the word "has" should probably be "have".

Author Response

Response to Reviewer 1 Comments

Point 1: The paper is interesting and I enjoyed reading it. The discussion relating to possible future developments in automotivity is topical and worthwhile. The study uses secondary data to good effect, although they have not really explained why this was the best choice of approach. The material used in the analysis is clearly set out and displayed.

Response 1: Thank you. The use of secondary data was mandated by the situation, in which we sought to understand the impact of COVID 19 as events were unfolding and being reported. In addition, the scope of the discussion is very broad. We also take in multiple aspects of ongoing change in automobility using a range of sources including the mainstream academic literature, and reports or other data from governments, agencies, companies, consultants, and others. Rather, we adopt the methodological approach of longitudinal immersion [43]. It is not possible to cover this broad range though primary research. We have inserted this comment at the start of Section 3, page 6.

Point 2: Having said this, there are a number of areas on which I have concerns. Coming from a Systems background, I found the authors' use of Systems terms confusing. When siting work in a Systems paradigm, it is important to explain which systemic perspective is taken. The authors appear to have a hard, closed systems approach similar to that taken by, for instance, classical economics. However, even within such a perspective, it is necessary to recognize that boundaries are not given, but set by the person to whom the analysis is interesting. Boundaries do not 'disintegrate' as if they were a part of the physical world. They may become inappropriate and are redrawn. Similarly, the concept of a socio-technical system does not have one, universally-agreed definition for all purposes. The concept is used across many fields, e.g. business, geography, information systems, engineering and is defined by researchers in many different ways. The authors have given a definition but without setting the context within which they believe it to be valid, nor do they provide any references to confirm its acceptance and usefulness Lines 67-76). Furthermore, there is little material in this discourse that relates to systemic analysis or STS. The authors discuss transitions, mentioning particular systems at times. Perhaps systemic analysis is not necessary to their purpose. If that is the case, what is the relevant framework and literature?

Response 2: These are very important points. As the reviewer notes, there are various definitions and contextual operationalisations of the concept of socio-technical system. We have sought to clarify our usage of systems by reference to some of the early work on socio-technical transitions (Geels, 2002 etc.). We did not intend to convey a view of systems as being ‘hard’ or closed, and have sought therefore to provide a better explanation of our position. Rather as happens in the realm of industrial ecology (which draws on natural ecology to describe and analyse social and economic systems), we have viewed the systems concept as a metaphor rather than a reality. The reviewer is absolutely correct to identify that in defining the socio-technical system boundaries we are imposing a conceptual frame rather than reflecting a physical reality, even though the conceptual frame has a physical manifestation. In consequence, the reviewer is also absolutely correct to raise concerns over our use of the term ‘boundaries’. Again, our usage is metaphorical so that the reader can envisage the possibility of different ways that automobility might change. In this respect systemic analysis in the sense of, for example, causal loop diagrams, is not our precise purpose although the research in the future may extend into a more quantified phase. We have therefore given greater emphasis to our foundational literature in the socio-technical systems realm.

Inserted into top of section 2:

Socio-technical systems are comprised of coherent networks of production and consumption in which the key constituent elements are largely self-reinforcing. Systems in this sense are distinct from ‘hard’ or ‘closed’ systems as understood in the physical sciences. Rather, they are user-defined, albeit grounded in some concrete reality. At the core of socio-technical systems is a ‘regime’ of dominant technologies and associated social or economic structures. These structures include markets, companies, labour organisations, prevailing social norms and behaviours, legal frameworks, and government which act collectively to provide coherence to the system [10,11]. Socio-economic institutional entities are in this conceptualization also system components that mediate the agency of actors, and as such are intimately involved in the definition of system structure and change [11]. The domain of socio-technical transitions is then the focus of research and policy into enabling the transition of an existing socio-technical system into one that is more sustainable along one or more dimensions (see for example https://transitionsnetwork.org/). In a sense, the concept of systems used in this domain of research is an abstraction and simplification, or a metaphor as a means of assisting in the visualization of important but complex processes of change in society. Metaphors drawn from the physical sciences are often used in this manner, in for example the fields of evolutionary economics or of industrial ecology [12,13].

Also inserted into the top of Section 2:

That is, theoretically a socio-technical system has two important properties: it is reproduced by the internal relationships of the system components, and it has a defined conceptual boundary that can be used to confine the unit of analysis. The boundaries within and between systems are not entirely fixed, but have become of research interest as previously distinct socio-technical systems have come to coalesce in multi-system interactions [14,15].

Also inserted below Table 1:

Conceptually, this means that the socio-technical system can be considered as largely self-contained, and that the boundaries are impermeable. This means that there are limited interactions with other (adjacent) socio-technical systems, and new entrants are unable to penetrate the regime. A degree of technological and socio-economic convergence, particularly around digitization and electrification, has resulted an erosion in this boundary condition for systems that used to be considered separate. In multiple ways, there are thus varying degrees of ‘overlap’ between say automobility, housing, electricity, food, telecommunications, and so forth.

Also inserted below Table 1 (following paragraph):

Again, the boundary conditions defined in this paper can be considered as metaphors for complex societal interactions and processes that are both contested and uncertain in outcome.

Point 3:There are some sweeping statements in the Introduction that require evidence to support them. For example, on Line 77 "Generally, in STS research, the focus is on transition". Is this so? What is the field within which this is statement claims validity? Where are the references to support it? How are the authors defining 'transition'? Such references as there are in this section relate to sustainability, but not specifically to STS.

Response 3: Again, valid points that really reflect the desire to keep the paper compressed and ‘tight’. In essence, the focus is on sustainability transitions, which is the core interest of the annual conference (International Sustainable Transitions) of the global group of researchers in the Socio Technical Research Network (STRN) (see https://transitionsnetwork.org/). Hence, we did not regard the statement as problematic. However, to clarify our position we have located our analysis more precisely in the discourses of this group.

Inserted into L92:

Generally, in socio-technical system research the focus is on sustainable transitions (see Köhler et al., 2019).

(Note this reference is the key document for the future research agenda of the STRN, and outlines all the various empirical domains, theoretical and methodological approaches, new areas of research, etc. for this area of academic research.)

Point 4:"The role of a landscape ‘crisis’ event ... is not well developed in the literature". The literature of which field? Sustainability? Socio-technical Systems? What about the literature of Risk and Crisis Management? "more recent research has sought to understand ... " More recent than what? There is only one reference in this section and it relates to Energy research.

Response 4: Yes, this is an accurate comment. Our view is that in the realm of sustainability transitions in socio-technical systems the emphasis has been on long-run processes of change, often unfolding over decades, as technologies permeate society. Again, we agree we have under-referenced this key point, and now have sought to develop a statement with more clarity and references on the issue of temporality in such transitions. Of course, the reviewer is correct to identify that other literatures do indeed have more to say on the theme of crisis events. We have sought to focus the discussion, however, on crisis and socio-technical transitions.

We have changed the text to:

The role of a landscape ‘crisis’ event as a decisive historical moment that transforms a socio-technical system is not well developed in the Socio-Technical Transitions Network (STRN) literature, where the focus has been on the gradual unfolding of transition processes [30,31]. Again, some more recent research has sought to understand how transitions might be accelerated [32], in part because of the perception that sustainability problems were still growing, and planetary boundaries were in danger of being breached despite decades of policy intervention [33].

(Note: Kanda, W. and Kivimaa, P., 2020. [30] What opportunities could the COVID-19 outbreak offer for sustainability transitions research on electricity and mobility?. Energy Research & Social Science, 68, p.101666. is a recent paper that identifies this previous focus on long-term change processes. The paper by Wells et al. (2020) [31] does the same, but on the basis of using socio-technical transitions theory for futures orientated scenario analysis)

Inserted ahead of Table 1:

Again, the focus of transitions research has mostly been on technological innovation as the primary trigger to initiate system change [34], around which behaviours and attitudes, infrastructures, and institutions, might change with time. However, it has been recognised that the ‘lens’ of socio-technical transitions can contribute important insights into understanding the impact of the pandemic crisis event [30,31]. In part, the contribution of socio-technical transitions theory draws on the insights from research into the history of previous transitions in which a constellation of agents has acted as participants in system change processes wherein technological diffusion arises out of the co-construction of a technology in its contextual setting [35].

Point 5: I would like to see the authors reconsider what their objectives are in writing this paper and what is the real frame of reference for the research. A clearer statement of perspectives, including which bodies of theory are relevant, would be useful to the reader. The authors need to ensure that they have provided a complete set of references.

Response 5: We believe we have now provided a more robust academic foundation to the framing of this research, with the clarity desired and with a more complete set of references. In total the following references were added:

Geels, F. (2002). Technological transitions as evolutionary reconfiguration processes: A multi-level perspective and a case-study, Research Policy, 31(8-9), 1257-1274.

Ehrenfeld, J., 2004. Industrial ecology: a new field or only a metaphor?. Journal of cleaner production, 12(8-10), pp.825-831.

Chertow, M. and Ehrenfeld, J., 2012. Organizing self‐organizing systems: Toward a theory of industrial symbiosis. Journal of industrial ecology, 16(1), pp.13-27.

Geels, F.W., 2004. From sectoral systems of innovation to socio-technical systems: Insights about dynamics and change from sociology and institutional theory. Research policy, 33(6-7), pp.897-920.

Wells, P. and Nieuwenhuis, P. (2017) Operationalising deep structural sustainability in business: longitudinal immersion as extensive engaged scholarship, British Journal of Management, Vol. 28, 45–63.

Cohen, M.J., 2012. The future of automobile society: a socio-technical transitions perspective. Technology analysis & strategic management, 24(4), pp.377-390.

Argues that end of automobility is premature… lock in and path dependencies

Geels, F.W., 2018. Low-carbon transition via system reconfiguration? A socio-technical whole system analysis of passenger mobility in Great Britain (1990–2016). Energy research & social science, 46, pp.86-102.

Kanda, W. and Kivimaa, P., 2020. What opportunities could the COVID-19 outbreak offer for sustainability transitions research on electricity and mobility?. Energy Research & Social Science, 68, p.101666.

Rosenbloom, D., 2020. Engaging with multi-system interactions in sustainability transitions: a comment on the transitions research agenda. Environmental Innovation and Societal Transitions, 34, pp.336-340.

Geels, F.W., 2018. Disruption and low-carbon system transformation: Progress and new challenges in socio-technical transitions research and the Multi-Level Perspective. Energy Research & Social Science, 37, pp.224-231.

Kanger, L., Geels, F.W., Sovacool, B. and Schot, J., 2019. Technological diffusion as a process of societal embedding: Lessons from historical automobile transitions for future electric mobility. Transportation Research Part D: Transport and Environment, 71, pp.47-66.

Point 6: While the paper is generally well-written, it would benefit from another proof-reading. In particular, in Line 138, the word "on" should probably read "one"; on Line 159, the word "address" should probably be "addressing"; and at Line 163, the word "has" should probably be "have".

Response 6: Yes, good work! It is so hard to pick up all these minor errors. We have changed the items identified.

Reviewer 2 Report

This is an interesting and timely paper.  The paper is well written and flows beautifully, I would suggest additional references / sources for the tables, as they are binary at the moment, additional network diagrams would enhance the paper, visuals showing the links of the impact of covid-19 and the impact on the climate action agenda would be useful. I look forward to seeing how your work develops.

Author Response

Response to Reviewer 2 Comments

Point 1: This is an interesting and timely paper.  The paper is well written and flows beautifully, I would suggest additional references / sources for the tables, as they are binary at the moment.

Response 1:  Thank you for your comments. The references or sources for the tables are rather more difficult than at first would appear, if only because there are so many! Table 1 is a conceptual synthesis arising from the authors reflecting upon their work to date in this area to arrive at the framing so presented. Table 2 is especially challenging in this regard. We have therefore added more references to the text that discusses Table 2. The issue is this. Mostly, the academic or other sources focus on just one specific aspect of one issue (say for example, ethical aspects of the issue of autonomous cars). We have provided the news sources for Table 3.

SourcesTable3

ACEA (2019) Electric vehicles: tax benefits & incentives in the EU, European Automobile Manufacturers Association.

Hampel, C. (2020a) The Netherlands goes for EV purchase subsidies, https://www.electrive.com/2020/03/05/the-netherlands-goes-for-ev-purchase-subsidies/, Accessed 05/03/2020.

Randall, C. (2020a) EU Commission considers 100 billion euro transport package, https://www.electrive.com/2020/05/20/eu-commission-considers-100-billion-euro-transport-package/, Accessed 20/05/20.

Hampel, C. (2020b) France introduces electric car package, https://www.electrive.com/2020/05/27/france-presents-electric-car-package/, Accessed 27/05/20.

Randall, C. (2020b) Germany doubles EV subsidies, no more diesel support, https://www.electrive.com/2020/06/04/germany-doubles-ev-subsidies-no-more-diesel-support/, Accessed 04/06/2020.

Miller, C. (2020) Government offers subsidies only for electric vehicles and rejects scrappage scheme for petrol and diesel autos, https://www.ft.com/content/d409971c-791e-4602-b541-60c126d2e26e? Accessed 05/06/20.

Manthey, N. (2020) UK scrappage scheme and Tesla Giga Bristol rumoured, https://www.electrive.com/2020/06/08/uk-scrappage-scheme-and-tesla-giga-bristol-rumoured/, Accessed 08/06/2020.

Baggott, J. (2020) Fresh calls for a scrappage scheme or incentives to help boost new car demand, https://cardealermagazine.co.uk/publish/calls-scrappage-scheme-boost-new-car-demand-gather-pace/190981, Accessed 05/05/20.

Pickard, J. and Campbell, P. (2020) Car scrappage scheme to boost sales ‘unlikely’, say ministers, https://www.ft.com/content/3d3ff487-3cdc-4226-b912-2bdc4ecb491d?, Accessed 10/06/20.

Point 2: additional network diagrams would enhance the paper, visuals showing the links of the impact of covid-19 and the impact on the climate action agenda would be useful. I look forward to seeing how your work develops.

Response 2: A diagram is a very good idea. Here is the diagram and the discussion about it.

Figure 1: Crisis events and socio-technical transitions

Figure 1 illustrates schematically the impact of a crisis event on socio-technical transitions. Reading from left to right, the socio-technical system is assumed to be in a state of dynamic stability ahead of the crisis event. There is, however, also an assumption that alternative technologies, behaviours and practices are potentially available, while adjacent socio-technical systems are potentially developing towards the focal system. Hence, incipient transition pathways may be present. The crisis event then has the potential to accelerate or magnify change processes. First, it weakens the existing socio-technical regime. Markets may collapse, dominant incumbents are no longer profitable, and behaviours and practices can change rapidly. In parallel, the crisis may result in a failure of legitimacy for the dominant socio-technical system, precipitating a shift in government policies and a new openness to alternatives. Three broad outcomes are portrayed. The socio-technical system may become destabilised and collapse, it may recover and resume on a pathway of dynamic stability or it may undergo an accelerated transition to a new socio-technical system.

Round 2

Reviewer 1 Report

Thank you for your considered response and the additions and amendments you have made. This is a most interesting paper.

My one comment would be that, for the future, you need to consider your whole audience. Not all readers will have been aware of the conference from which the idea for your paper arose. Good luck with the next stages of your research.